# Outcomes and Patterns of Evolution of Patients with Hematological Malignancies during the COVID-19 Pandemic: A Nationwide Study (2020–2022)

**DOI:** 10.3390/jcm13185400

**Published:** 2024-09-12

**Authors:** Rafael Garcia-Carretero, Maria Ordoñez-Garcia, Ruth Gil-Prieto, Angel Gil-de-Miguel

**Affiliations:** 1Internal Medicine Department, Mostoles University Hospital, Rey Juan Carlos University, 29835 Mostoles, Spain; 2Hematology Department, Mostoles University Hospital, Rey Juan Carlos University, 29835 Mostoles, Spain; 3Department of Preventive Medicine and Public Health, Rey Juan Carlos University, 28922 Alcorcón, Spain; ruth.gil@urjc.es (R.G.-P.); angel.gil@urjc.es (A.G.-d.-M.)

**Keywords:** SARS-CoV-2, COVID-19, hematologic malignancy

## Abstract

**Background**: Early reports suggest that hematological malignancy (HM) is a relevant risk factor for morbidity and mortality in COVID-19. We investigated the characteristics, outcomes, and risk factors for mortality in patients hospitalized with HM and COVID-19. **Methods**: We conducted a retrospective, nationwide study using data from hospitalized patients that were provided by the Spanish Ministry of Health including all patients admitted to a Spanish hospital from 2020 to 2022 with a COVID-19 diagnosis. A descriptive analysis and correlational analyses were conducted. Logistic regression was used to assess mortality in these patients and to calculate odds ratios (ORs). **Results**: We collected data on 1.2 million patients with COVID-19, including 34,962 patients with HMs. The incidence of hospitalization for patients with HMs was 5.8%, and the overall mortality rate was higher than for patients without HMs (19.8% versus 12.7%, *p* = 0.001). Mortality rates were higher for patients with lymphomas, multiple myelomas, and leukemias than for those with myeloproliferative disorders. Having HMs was a risk factor for mortality, with OR = 1.7 (95% CI 1.66–1.75, *p* = 0.001). By subtype, non-Hodgkin lymphomas were the highest risk factor for mortality (OR = 1.7), followed by leukemias (OR = 1.6), Hodgkin lymphomas (OR = 1.58), and plasma cell dyscrasias (OR = 1.24). **Conclusions**: This study identifies the different clinical profiles of patients with HMs who are at a high risk for mortality when hospitalized with COVID-19. As members of a vulnerable population, these patients deserve special prophylactic and therapeutic measures to minimize the effects of SARS-CoV-2 infection.

## 1. Introduction

Coronavirus disease 2019 (COVID-19) is caused by severe acute respiratory syndrome coronavirus 2 (SARS-CoV-2), identified in late 2019 [1,2]. Over the course of the pandemic, several variants of the virus presented modified transmissibility or immune escape from previous immunization [3,4]. The COVID-19 pandemic had a significant impact on the world, with 6.7 million confirmed deaths by December 2022 [5]. Currently, the disease is now considered an endemic, involving decreasing severity and mortality in the general population.

Patients with hematological malignancies (HMs) are at high risk for mortality from COVID-19 and are more susceptible to poor prognoses [6]. The clinical course of COVID-19 in patients with HM can be severe, as has been reported in several meta-analyses, with higher rates of hospitalization and higher intensive care unit (ICU) admissions [7,8,9]. In addition, the case fatality rate is higher in patients who have HMs [6,10]. It has been noted that the mortality rate associated with COVID-19 can reach 34%, although some reviews have reported mortalities up to 51% [7], or 3.5 times the rate in the general population [11].

Patients with HMs have a significantly reduced immune response to COVID-19, either due to the underlying disease itself or to the use of treatments that induce immunosuppression. Using vaccines to induce immunization in such patients could prevent a significant impact of COVID-19, but only half of patients develop a measurable amount of antibodies against SARS-CoV-2 [6].

Apart from the profound immune dysfunction in these patients, several risk factors for mortality have been identified, such as age, cardiovascular comorbidities, and HM subtype [12,13]. However, mortality in HM patients has reportedly been declining since the beginning of the pandemic. Moreover, the demographic and clinical profiles of the patients have changed over time, likely due to vaccination, the breakthrough of new variants of SARS-CoV-2, preventive measures, and effective treatments [9,11,14]. These events may have played a role in the changes in mortality.

HMs, either in the form of solid tumors or hematological neoplasms, are often under-reported, as series for hematologic patients are scarce or heterogeneous. Patients with HMs are more vulnerable and deserve special attention, particularly in cases of COVID-19. An improved understanding of the demographic features, clinical profiles, and outcomes of specific populations can improve the management of this population. It should be noted that the future of the evolving endemic remains uncertain, and studies analyzing the epidemiology and risks factors of patients with HMs can offer valuable insight into the interplay between HMs and COVID-19.

This study describes the clinical impact of hospitalization for COVID-19 in patients with HMs. We designed a retrospective, population-based, nationwide study among hospitalized COVID-19 patients with HMs to assess their clinical characteristics and mortality relative to COVID-19 patients without HMs. We also evaluated the predictors associated with mortality in COVID-19 patients with HMs.

## 2. Materials and Methods

### 2.1. Study Design and Data Collection

We conducted a population-based, retrospective study using microdata extracted from the National Hospital Data Information System at Hospitalization (MBDS-H) from the Spanish Ministry of Health between the years 2020 and 2022. This registry includes patient data and their diseases, encoded using the 10th Clinical Revision of the International Classification of Diseases (ICD-10-CM). We used the coding for COVID-19 (U07.1) in any diagnostic position (either main or secondary diagnosis) when collecting data.

MBDS-H is an administrative registry that is constructed from discharge reports covering nearly 95% of hospitals in Spain, both public and private. It is estimated that 97% of all discharge reports are collected in this database. The registry includes age, sex, date of admission/discharge, type of hospital, place of residence, and diagnoses at discharge. A new dataset is generated in January of each year. Due to the large amount of data, the data availability has a delay of 1 year. At present, the Spanish Ministry of Health has provided us with data up to 31 December 2022.

In addition to the confirmed diagnosis of COVID-19, patients were categorized using codes to include different types of lymphoma (Hodgkin, follicular, B-cell, T-cell, and NK-cell lymphoma), multiple myeloma and plasma cell leukemia, acute and chronic leukemias (lymphoid, myeloid, and monocytic), and myeloproliferative disorders (polycythemia vera, essential thrombocytosis, myelofibrosis, and chronic myelomonocytic leukemia), which are listed in Appendix A Table A1. We collected data on age, sex, date of admission and discharge, intensive care unit (ICU) admission, and death (if it occurred) for each hospitalized patient. Some chronic conditions of interest were also collected, such as diabetes, hypertension, and other comorbidities. Patients with incomplete data regarding ICU admission, mortality, length of hospitalization, and diagnosed conditions of interest were excluded. No names or personal information were recorded, and data were de-identified to ensure patients’ privacy. No data on treatment or immunological status were provided. This study was approved by the Research Committee of our institution.

### 2.2. Definition of Waves

The observation period covered 3 years (January 2020 to December 2022) and was split into several periods according to local epidemic outbreaks. We based the categorization of the pandemic on the classification of the epidemiological research by Epidemiological National Surveillance Net, which exclusively used data from Spain. Epidemic waves were based on 14-day cumulative incidence. Each turning point indicated the end of one wave and the beginning of the next [15]. Appendix A Table A2 presents the dates that we used to define the epidemic waves.

### 2.3. Statistical Analyses

We performed descriptive and correlational analyses. We calculated medians and interquartile ranges (IQRs) for continuous variables and absolute numbers and percentages for categorical variables. Hospital length of stay was defined as the total number of days of stay divided by the total number of hospitalizations and expressed as medians and IQRs. Deaths and ICU admissions were divided by the total number of hospitalizations to calculate the mortality rate and ICU admission rate, respectively. Both of these parameters were expressed as absolute numbers and percentages. Chi-square and Mann–Whitney U tests were performed as tests of independence wherever appropriate. In multivariate analysis, we performed binary logistic regressions to assess the effects of explanatory variables on mortality (considered the response variable). We calculated the regression coefficients and odds ratios (ORs) of the variables of interest.

For all tests, the level of statistical significance was set at *p* < 0.05. We used Python language version 3.11.2 and R language version 4.3.2 (Vienna, Austria) on a Debian 12 GNU/Linux workstation.

## 3. Results

### 3.1. Descriptive Analyses

The first known COVID-19 case in Spain was detected in late January 2020, but the first significant hospitalization of a patient due to COVID-19 occurred in early February 2020, coinciding with the spread of the virus in European countries like Italy. Spain’s first confirmed death from COVID-19, which was retrospectively identified, happened on 13 February 2020. Between 1 February 2020 and 31 December 2022, almost 1.2 million patients were hospitalized due to COVID-19 (Table 1). The cohort of patients with HMs included 34,962 individuals (2.94% of patients hospitalized with COVID-19). It was estimated by the Spanish Oncology Society [16] that there were 198,507 patients with HMs in Spain in 2022. That is, the prevalence of HMs in Spain was estimated to be 0.41%. The rate of hospitalization of patients in the general population was 0.83% per year between 2020 and 2022. For individuals with HM, we calculated an estimated ratio for hospitalized patients of 5.87% per year, that is, greater than the general population.

Men were more likely to be admitted to the hospital (54.9% globally). The median age of patients with HM was 75, and HM patients had a longer hospital stay than patients without HM (13.8 vs. 10.8 days). The ICU admission rate and mortality rate were also higher in patients with HM (9.8% and 19.8%, respectively). Additionally, we analyzed some comorbidities, including diabetes, hypertension, coronary disease, and so on. Table 1 presents the demographic and clinical features of our population. It should be noted that the prevalence of solid malignancies was lower in patients with HM.

We split the whole observation period into seven epidemic waves, as shown in Appendix A Table A2. The daily admissions are shown in Figure A1, while the monthly evolution of the pandemic during the 3 years of observation is shown in Figure 1. The evolution of mortality is shown in Figure 2. The admission and mortality trends match. As shown in Appendix A Table A3, in-hospital mortality reached its highest values in January 2021 and January 2022 (up to 26.6% in the third period) and its lowest value in December 2022 (14.7%). In spite of the overall decreasing mortality rate, cumulative deaths were 6925, with almost 1000 of them having occurred by May 2020 (Figure A2).

We also explored the relationship between age and sex in a population pyramid (Figure 3). The main characteristics related to sex among individuals with HM are given in Table 2. Admission was more likely among men (59.4%). The mortality rate was similar between men and women (20.1% vs. 19.3%). In addition, the prevalences of diabetes, coronary disease, chronic kidney disease, solid malignancies, and chronic pulmonary diseases were higher in men, while hypertension, heart failure, and obesity were more prevalent in women. The rate of ICU admission was higher in men than in women (10.6% vs. 8.6%).

### 3.2. Comorbidities

The subgroup of patients with myeloproliferative disorders had the highest prevalence of diabetes, coronary disease, heart failure, dementia, chronic kidney disease, chronic pulmonary disease, and cerebrovascular disease (Table 3).

### 3.3. Overall and Subtype-Related Mortality

According to the Spanish Oncology Society [16], the hospitalization rates per year were 1% (Hodgkin lymphoma), 3.7% (non-Hodgkin lymphoma), 12.6% (multiple myeloma), and 7.9% (leukemias). No data were available on chronic myeloproliferative disorders. Table 3 presents the characteristics of our population by subtype of HM.

Mortality rates were higher in patients with lymphomas, multiple myelomas, and leukemias than in those with myeloproliferative disorders. In-hospital mortality rates are plotted in Figure 4 and Figure A3. Overall, we observed a decline in mortality for all types of HM over time. Hodgkin lymphoma maintained a certain stability over the 3 years of observation, while the remaining malignancies settled down below 15% of mortality by the end of 2022 (see Appendix A Table A4).

### 3.4. Age-Adjusted Mortality

We also calculated the crude and age-standardized (adjusted) mortality rates using the direct method, as shown in Figure 5 and Appendix A Table A5. The age groups remained stable across the 3 years of the observation period. Figure 5 shows that, in 2022, patients over 60 experienced an increase in mortality rate, but no significant changes were observed in the rest of the groups.

### 3.5. Multivariate Analyses

We performed logistic regression to assess the effect of HM on mortality due to COVID-19 in the general population. The unadjusted OR was 1.7 (95%CI 1.66–1.75, *p* = 0.001), that is, hospitalized patients with HM had a 70% greater chance of dying due to COVID-19. Adjusted by other estimators (i.e., sex, age, and comorbidities), the OR did not change, as shown in Table 4.

Next, we performed a multivariate analysis of the cohort of patients with HM, that is, we explored the effects of several predictors in this specific population (Table 5). The model applying binary logistic regression identified non-Hodgkin lymphomas as major risk factors for mortality, followed by leukemias, Hodgkin lymphomas, and plasma cell dyscrasias. The model also evaluated the effects of age, sex, and included comorbidities. ICU admission and length of ICU stay were considered predictors of mortality, as they can increase the risk of in-hospital death.

## 4. Discussion

We report the first population-based, nationwide study of the epidemiology and the risk of in-hospital death in patients with prevalent HMs and COVID-19 in Spain. We included almost all patients hospitalized with COVID-19 from 2020 to 2022. Our aim was to gain insight into the characteristics of this specific population. The first concern that may arise regards the high rate of hospitalization (5.8%) in patients with HM relative to the general population (0.84%), which indicates the impact of infection with the SARS-CoV-2 virus in this specific population. The overall mortality rate of patients with HM in our study was 19.8%. We also found that those with male sex and elderly age had a significant risk for hospitalization, in line with other studies [11,14]. Demographic profiles in terms of sex, age, and comorbidities in patients with HM were similar to those found in the general population.

### 4.1. ICU Admissions

We identified a decreased ICU admission rate (9.8%) relative to early meta-analyses [7,17], which reported a pooled ICU admission of 21%. A meta-analysis by Langerbeins et al. [7] included studies of patients with COVID-19 and HM during the early stages of the pandemic. The researchers collected data from retrospective cohorts, prospective registries, and population surveys, identifying an ICU admission rate between 10% and 24%. We cannot explain these discrepancies among studies. One hypothesis could be that those meta-analyses reported data from the earliest stages of the pandemic, when ICU settings were overwhelmed and the disease was at its most severe. However, we observed increasing rates of ICU admissions over the analyzed period, with the lowest rate during the earliest stages. Moreover, in our cohort, we did not find rates above 14% in our analysis of the observation period by waves. Another concern that may arise regards the convenience of transferring the more severe patients with HM to these clinical areas. The percentage found in our study (9.8%) could not represent the severity of patients with HM. In fact, in an overwhelmed ICU setting, patients with any malignancy may have been denied admittance to these units. Therefore, our figures for ICU admissions could under-represent the severity of COVID-19 in this population. Unfortunately, we do not have data on the criteria for ICU admissions for our patients.

### 4.2. Overall Mortality

Early reports have reported different outcomes for patients with HM. A meta-analysis reported a mortality rate between 14% and 51% [7]. A Turkish study [18] estimated the risk of death in patients with HM at 14% compared to the general population (7%), which is similar to our results. However, overall, the outcomes and conclusions differ from ours. The significant disparity in the range of mortality across studies cannot be easily explained. The cited studies were based on large cohorts of patients with HM, but some were heterogeneous [7]. Some studies have reported rates of mortality in both the ambulatory and hospitalized population, while others have reported data from patients only with HM [12]. Another plausible explanation is that, similar to the results for ICU admission, these other studies and meta-analyses were published during the early stages of the pandemic, when mortality was the highest and ICU admissions were reserved for selected patients only.

A Spanish study of 1166 hospitalized patients with HM reported an overall mortality rate of 32% [19]. This rate is quite different from the rate we found (19.8% in patients with HM). This discrepancy may be explained by the difference in observation periods and the constrained area where the other study was conducted. That study only considered patients in Madrid (with a population of 6.6 million, 14% of the total Spanish population) over the first 12 months of the pandemic, before vaccination was widely available. It should also be noted that the region of Madrid was the most affected by COVID-19 during the first year of the pandemic [11], which might explain the high mortality rate.

We reported a noticeable increase in COVID-19-related mortality in Spain during late 2022, although it was milder compared to earlier waves. Several factors may have contributed to this increase. We can speculate about some factors. Omicron subvariants like BA.4, BA.5, and later BA.2.75 were circulating in Europe, including Spain. There were more immune-evasive. Although they caused mild disease in healthy individuals, these strains could still lead to severe outcomes in vulnerable populations like the elderly or immunocompromised. Immunity waning can be also an interesting factor. By late 2022 most of the population in Spain had been vaccinated, but their immunity might have waned. The reduced immunity against SARS-CoV-2 might have contributed to increase mortality during this period. Another factor could be the impact on vulnerable populations. The elderly, the immunocompromised, and those patients with comorbidities might have been at a high risk even during mild peaks. Also, the co-circulation of influenza and Respiratory Syncytial Virus in late 2022, alongside SARS-CoV-2, could have contributed to the increase in mortality. So, although merely speculation, we cannot attribute the increase in mortality solely to the variants but to a more complex interplay of the mentioned factors.

### 4.3. Mortality in Subtypes of Malignancy

According to Vijenthira et al. [17], who examined 2192 patients in a meta-analysis, the pooled risk for mortality was 41% for leukemias, 33% for plasma cell dyscrasias, 32% for lymphoma, and 34% for myeloproliferative disorders. The aforementioned Spanish study reported that the most common subtypes were non-Hodgkin lymphoma and multiple myeloma [19]. Another Spanish study that included immunosuppressed patients found the highest risk of mortality in patients with leukemia and lymphoma [13], although no subtype of malignancy was reported. Overall, the results for mortality in these studies are higher than ours. Here as well, we consider the heterogeneity of the given studies as the cause of the difference between their results and ours.

The most relevant study that examined patients with HM infected by SARS-CoV-2 is EPICOVIDEHA [6,12], a survey supported by the European Hematology Association (EHA). EPICOVIDEHA found that non-Hodgkin lymphoma (31%) and plasma cell disorders (17%) were the most prevalent. That study also found that leukemias and myelodysplastic syndrome had the highest mortality rate (in a range from 18% to 21%), in line with our study. EPICOVIDEHA reported an overall decrease in mortality over time and across all subtypes and a steady decline in mortality per type of malignancy, from a global 38.2% to 5.3% in late December 2022. While these results are consistent with the global trends found in our study, our results are less optimistic. We identified a decline in mortality across all subtypes of HM, beginning in the final quarter of 2021, producing an overall rate of 15% in late December 2022. However, during that period, only multiple myeloma and plasma cell dyscrasias showed a rate as low as 8.5%, with the remaining subtypes being above 13%, and Hodgkin lymphoma having a 17.5% mortality rate. That is, our reported mortality rate is higher than the results from EPICOVIDEHA. In spite of these discrepancies, the results are consistent with an overall decline in mortality rate in Spain [11], which was probably concurrent with the beginning of vaccination rollout in patients with malignancies in Spain in March 2021. Unfortunately, we have no data regarding the role of vaccination in patients with HM, and research on this topic is purely speculative.

### 4.4. Comorbidities and Predictors of Mortality

We observed that the prevalence of comorbidities in COVID-19 patients with HM did not differ from that of the general population. It should be noted that solid malignancies were more prevalent in patients without HM. We also found that the subgroup with myeloproliferative disorders had the highest prevalence of cardiovascular comorbidities such as cerebrovascular disease, diabetes, coronary disease, and heart failure, as well as dementia, chronic kidney disease, and chronic pulmonary disease, probably because this subgroup includes an older population that is vulnerable to chronic diseases. From 60 years old and older, we observed an increase in deaths, indicating that mortality is strongly associated with age.

The interest in comorbidities is not only descriptive but also predictive. Some studies have associated certain conditions with poor prognosis. The meta-analysis by Langerbeins et al. [7] summarized the main risk factors for severity. Among comorbidities, the authors identified demographic features, such as older age and male sex, as well as several comorbidities, such as cardiovascular features and chronic diseases. The meta-analysis reported factors such as race, cytotoxicity, anti-cancer therapy, and type of malignancy. For HM, lymphomas were the most important predictive marker related to mortality. Regalado et al. [20] found that older age, heart disease, and chronic kidney disease were the most important risk factors for mortality in a multivariate analysis of patients with lymphoma.

The EPICOVIDEHA registry [6] found that cardiomyopathy (35%), diabetes (14%), and chronic pulmonary disease (13%) were the most common conditions in patients with HM. It also reported several predictors associated with mortality using Cox regression and hazard ratios (HRs) as follows: age (HR 1.03), two or more comorbidities (HR 1.24), and non-HM (HR 1.83). Chronic cardiomyopathy, chronic liver disease, and chronic kidney disease were the most relevant predictors in a multivariate analysis assessing mortality [12].

Chronic pulmonary obstructive disease (COPD) is commonly associated with smoking. Interestingly, the EPICOVIDEHA registry found that chronic pulmonary disease was frequent among patients with HM, but it was not related to higher mortality. This is surprising, as COPD is generally considered a risk factor for severe COVID-19, where an increased odds ratio (OR) would be expected. Several factors may explain this lower-than-expected mortality, but they are purely speculative. First, COPD patients are likely to receive closer monitoring and earlier therapeutic interventions, potentially reducing the severity of COVID-19. Additionally, respiratory therapies, such as bronchodilators or corticosteroids, could help manage symptoms early on. The anti-inflammatory effects of corticosteroids may also lower the risk of the hyperinflammatory *cytokine storm* associated with severe COVID-19. Survivor bias might play a role, as HM patients with COPD may be younger or better managed, making them more resilient. Furthermore, surviving the early pandemic waves could have provided these patients with better care protocols or timely vaccinations, reducing their mortality risk.

### 4.5. Limitations

Our study had several limitations. Its main limitation was the use of an administrative database. Although electronic health records help researchers collect data and make them available for research, some clinical data were not available. Information on vaccination and medications was not available, so we could not assess the effect of immunization or certain treatments, such as specific anti-cancer treatments or immunotherapies. As a result, the clinical stage of the patients could not be evaluated, so we could not assess risk stratification in patients with HM. This issue is inherent to the characteristics of this dataset and the coding process for MBDS-H.

In addition, some patients’ data may have been duplicated. The database provided by the Ministry of Health is anonymized by default, so we could not identify duplicated patient data. It is important to remember that patients with HM tend to have frequent medical visits; thus, each entry of data in our study would represent one given hospitalization rather than one given patient. While this issue may result in bias, every hospitalization is unique. We consider that overestimation is unlikely, due to the large number of patients included.

We included hospitalized patients, so another limitation is that we did not include ambulatory patients. This may have led our outcomes to under-represent reality.

Another limitation is that we assumed that mortality was directly related to COVID-19. However, some HMs are life-threatening on their own. We are aware that COVID-19 can be a healthcare-associated infection, and was particularly so in later 2022. In fact, other authors have reported that in-hospital mortality in patients with HM may increase by 50% if it overlaps with COVID-19 infection [21,22].

### 4.6. Strengths

Regardless of these limitations, we consider that the main strength of our study is its inclusion of almost all hospitalized patients with COVID-19 and HM, such that our results (ICU admission of 9.8% and mortality rate of 19.8%) are reliable and capture the impact of COVID-19 in this specific population. Our study represents the largest set of data on hospitalized patients with HM and COVID-19. We are aware of the heterogeneity of results across different studies, but this can be explained by the bias introduced to a given study according to the specific population considered in that study. By conducting a Spain-wide study, we reliably assessed the risk of in-hospital mortality in this patient group based on data from an unrestricted population in terms of place or region.

Our findings may be useful for understanding the impact of COVID-19 in patients with HM. Meta-analyses and nationwide studies such as ours may be very relevant because they analyze large cohorts and can identify trends across specific populations. Collecting knowledge and analyzing data on the impact of COVID-19 in specific at-risk groups is critical for the endemic phase of COVID-19. Our reported outcomes highlight the burden of healthcare in patients with HM. Future prevention strategies should emphasize this population. Moreover, patients with HM should be considered for ICU admission when appropriate, given that they can recover from COVID-19 [6].

Early prophylactic or therapeutic measures should be promptly initiated in such patients to prevent severe forms of COVID-19. Our study demonstrates the vulnerability of patients with HM. We suggest that treatment for COVID should not be delayed. Physicians should also be aware of the need for close monitoring and tailored care for these patients.

## 5. Conclusions

This study presents the first comprehensive, nationwide analysis of the epidemiology and risk factors for in-hospital mortality among patients with hematological malignancies (HMs) and COVID-19 in Spain. It highlights the unique vulnerabilities of this population. Our findings indicate a significantly higher hospitalization rate (5.8%) for patients with HM compared to the general population (0.84%), and an overall mortality rate of 19.8%.

We found discrepancies in mortality rates between our research and other studies, potentially due to differences in study populations, observation periods, and pandemic phases. Notably, the mortality rate has declined over time, possibly reflecting improved management and vaccination, although we are aware that specific data on vaccination were not available in this study.

Our research highlights the need for targeted prophylactic and therapeutic interventions. Close monitoring and tailored care are essential to mitigate the severe outcomes of COVID-19 in this high-risk population. Our study provides valuable insights into the impact of COVID-19 on patients with HM in a population-based setting.

COVID-19 has turned from a pandemic to an endemic, and it will pose future healthcare challenges in both general and specific populations. This is why a better understanding will benefit the creation of optimized strategies for patients with HM. Although the severity of the disease and mortality have decreased over time, patients who have HM are at high risk, as the impact of COVID-19 remains high. Our study highlights the importance of preventive measures, targeted interventions, vaccination, and prompt therapy to improve survival and prevent severe forms of COVID-19.

## Figures and Tables

**Figure 1 jcm-13-05400-f001:**
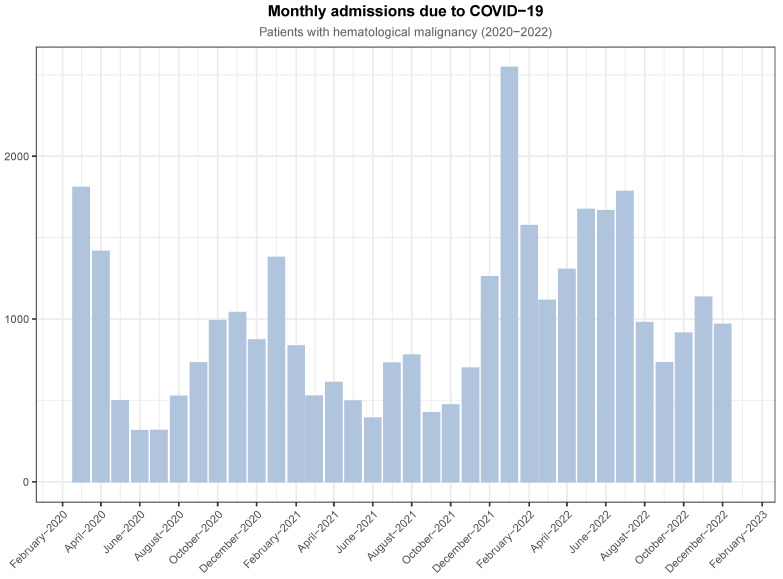
Evolution and trend of the COVID-19 pandemic with respect to hospitalized patients with hematological malignancies over the studied period, aggregated by month.

**Figure 2 jcm-13-05400-f002:**
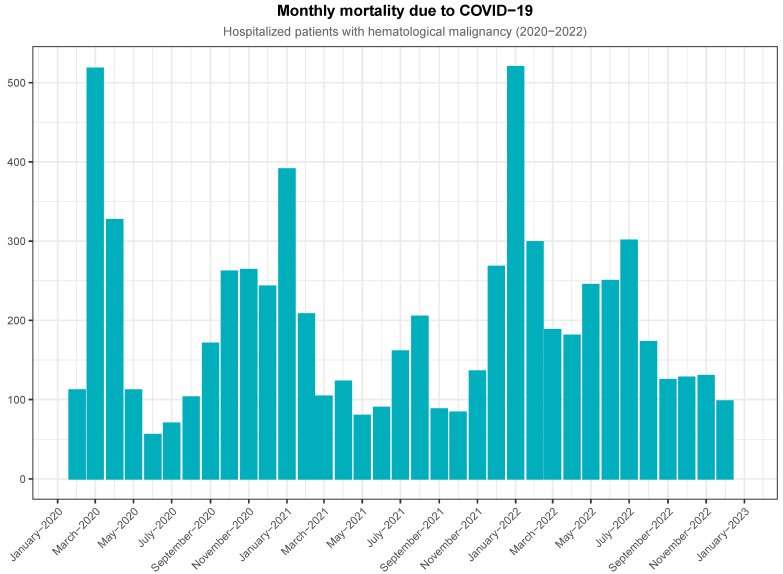
In-hospital mortality of patients with COVID-19 and hematological malignancy.

**Figure 3 jcm-13-05400-f003:**
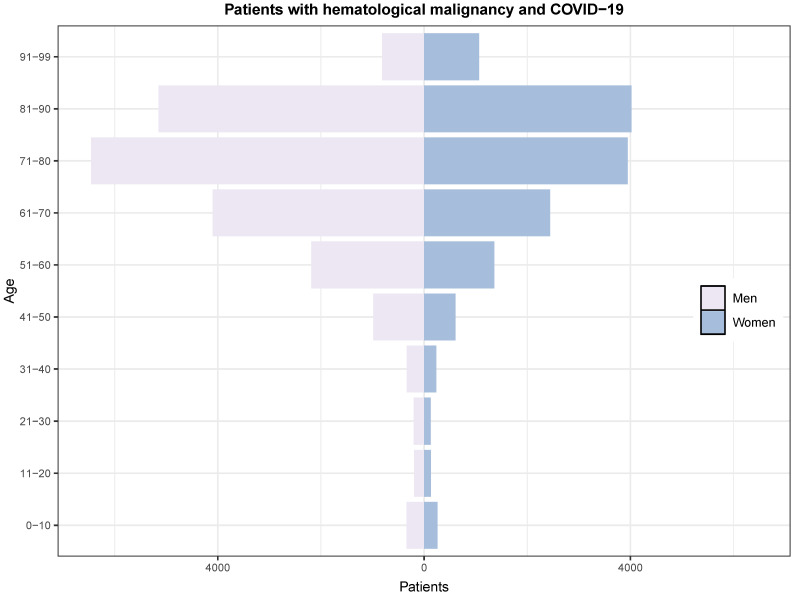
Population pyramid showing the distribution according to sex and age between 2020 and 2022.

**Figure 4 jcm-13-05400-f004:**
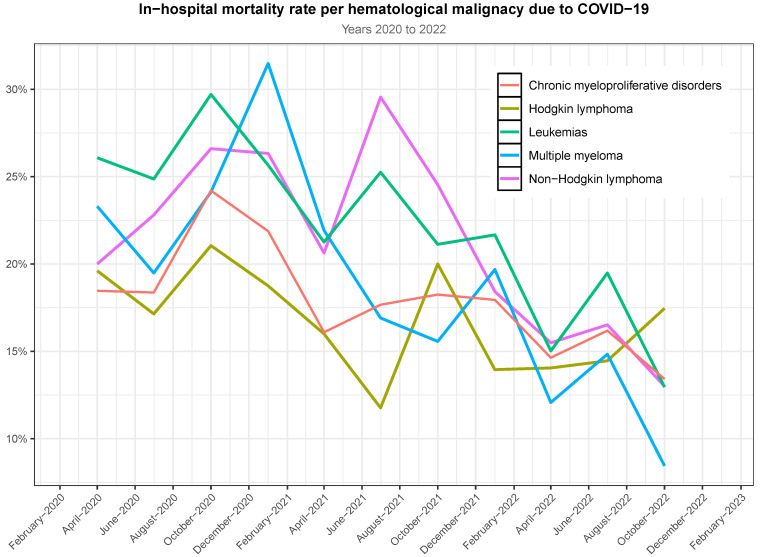
Mortality rates per type of malignancy. Non-Hodgkin lymphomas are a combination of follicular, B-cell, and T/NK-cell lymphomas. Multiple myeloma also includes plasma cell neoplasms.

**Figure 5 jcm-13-05400-f005:**
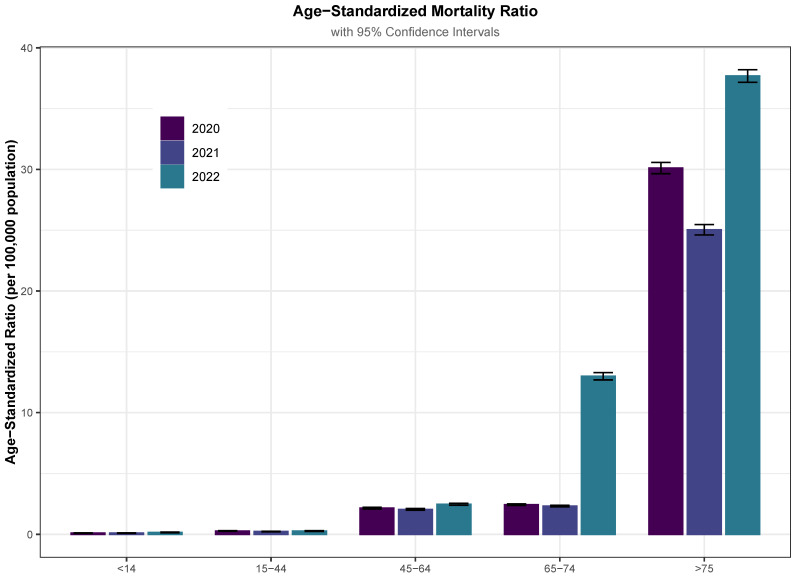
Age-standardized (adjusted) mortality rate using the direct method.

**Table 1 jcm-13-05400-t001:** Demographic and clinical characteristics of patients in our cohort.

	Patients with HM	Patients without HM	All Patients	*p*-Value
Patients	34,962	1,153,198	1,188,160	NA
Sex (men, %)	59.4%	54.8%	54.9%	0.001
Age (IQR)	75 (19)	73 (27)	73 (27)	0.001
Hospital LOS in days (IQR)	13.8 (11)	10.8 (8)	10.9 (8)	0.001
ICU admissions	3420	97,158	100,578	NA
ICU (%)	9.8%	8.4%	8.5%	0.001
ICU LOS in days (IQR)	8.5 (17.2)	8 (20.3)	8 (20.2)	0.598
Deaths	6925	146,219	153,144	NA
Mortality rate (%)	19.8%	12.7%	12.9%	0.001
Comorbidities				
Diabetes	22.2%	24.5%	24.4%	0.001
Hypertension	32%	31.9%	31.9%	0.805
Coronary disease	8.6%	9.8%	9.7%	0.001
Heart failure	15.9%	17%	17%	0.001
Dementia	3.9%	6.6%	6.5%	0.001
Chronic kidney disease	17%	15.5%	15.6%	0.001
Chronic liver disease	0.7%	0.6%	0.6%	0.017
Solid tumor	5.8%	7.4%	7.4%	0.001
Obesity	7.1%	12%	11.8%	0.001
Chronic pulmonary disease	12.7%	14.8%	14.8%	0.001
Cerebrovascular disease	0.8%	1.4%	1.4%	0.001

Categorical variables are expressed in absolute numbers and percentages. Continuous variables are expressed as medians (interquartile ranges). Chi-square test for continuous variables and Mann–Whitney U test for categorical variables were performed as tests of independence. NA: not applicable. ICU: intensive care unit. IQR: interquartile range. HM: hematological malignancy. LOS: length of stay.

**Table 2 jcm-13-05400-t002:** Demographic and clinical characteristics of the studied cohort of patients with hematological malignancy.

	Men	Women	Total	*p* Value
No. patients	(*n* = 20,764)	(*n* = 14,198)	(*n* = 34,962)	NA
Age, years (median, IQR)	74 (19)	76 (19)	75 (19)	0.001
Hospital LOS in days (median, IQR)	14 (12)	13.5 (11)	13.8 (11)	0.005
ICU admissions	2194	1226	3420	NA
ICU (%)	10.6	8.6	9.8	0.001
ICU LOS in days (median, IQR)	9 (17.6)	8 (16.2)	8.5 (17.2)	0.028
Deaths	4182	2743	6925	NA
Mortality rate (%)	20.1%	19.3%	19.8%	0.008
Comorbidities				
Diabetes	23.7%	20%	22.2%	0.001
Hypertension	31.4%	32.8%	32%	0.003
Coronary disease	11%	5.1%	8.6%	0.001
Heart failure	14.5%	17.8%	15.9%	0.001
Dementia	3.2%	5%	3.9%	0.001
Chronic kidney disease	18.1%	15.4%	17%	0.001
Chronic liver disease	0.8%	0.6%	0.7%	0.037
Solid tumor	7%	4%	5.8%	0.001
Obesity	6.2%	8.4%	7.1%	0.001
Chronic pulmonary disease	17.6%	5.5%	12.7%	0.001
Cerebrovascular disease	0.8%	0.9%	0.8%	0.457

Categorical variables are expressed in absolute numbers and percentages. Continuous variables are expressed as medians (interquartile ranges). Chi-square test for continuous variables and Mann–Whitney U test for categorical variables were performed as tests of independence. NA: not applicable. ICU: intensive care unit. IQR: interquartile range. HM: hematological malignancy. LOS: length of stay.

**Table 3 jcm-13-05400-t003:** Incident cases, deaths, and demographic characteristics of hospitalized patients with hematological malignancy.

	Hodgkin Lymphoma	Follicular and B-Cell Lymphoma	T/NK-Cell Lymphomas	Multiple Myeloma and Malignant Plasma Cell Neoplasms	Leukemias	Chronic Myeloproliferative Disorders
No. patients	860	10,482	766	6171	12,628	5308
Adjusted hospitalization rate per year	1%	3.7% *		12.6%	7.9%	ND
Sex (men, %)	64.4	59.1	64.8	57.3	61.4	58.6
Age, years (median, IQR)	61 (32.2)	73 (18)	69 ( 21)	75 (16)	76 (19)	79 (18)
Hospital length of stay in days (median, IQR)	14.3 (12.2)	14.7 (13)	15.4 (13)	12.7 (10)	14 (12)	12.3 (9)
ICU admissions	100	1111	90	499	1242	483
ICU (%)	11.6	10.6	11.7	8.1	9.8	9.1
ICU length of stay in days (median, IQR)	6 (12)	9 (18)	9.5 (19)	7 (15)	9 (16)	9 (15)
Deaths	143	2104	155	1123	2757	976
Mortality rate (%)	16.6%	20.1%	20.2%	18.2%	21.8%	18.4%
Comorbidities						
Diabetes	17.4%	21.5%	20.9%	20.7%	23.7%	24.2%
Hypertension	22.1%	32.2%	31.3%	33.8%	31.7%	31.3%
Coronary disease	5.5%	7.3%	6.5%	8.2%	9.2%	12.1%
Heart failure	8.8%	12.2%	12.5%	17.6%	16.4%	23.8%
Dementia	1.5%	3%	2.5%	3.8%	3.9%	6.7%
Chronic kidney disease	8.4%	13.1%	10.7%	23.4%	16.2%	23.4%
Chronic liver disease	1.2%	1%	0.4%	0.6%	0.5	0.7%
Solid tumor	4.5%	6.6%	6.7%	5.2%	5.1%	6.7%
Obesity	6.4%	7.1%	6.3	7.0	7.0	7.9
Chronic pulmonary disease	11.6%	11.9%	11.7%	12.7%	12.0%	17.6%
Cerebrovascular disease	0.5%	0.5%	0.5%	0.8%	0.9%	1.3%

Categorical variables are expressed as absolute numbers and percentages. Continuous variables are expressed as medians (interquartile ranges). NA: not applicable. ICU: intensive care unit. IQR: interquartile range. ND: nonavailable data. LOS: length of stay. * Data for non-Hodgkin lymphoma are combined data from follicular, B-cell, and T/NK-cell lymphomas.

**Table 4 jcm-13-05400-t004:** Multivariate analyses (logistic regression) for the entire cohort using mortality as the dependent variable.

	OR (Adjusted)	95%CI	*p* Value
Sex (men)	1.35	1.34–1.37	0.001
Age	1.05	1.05–1.05	0.001
Hematological malignancy	1.71	1.66–1.76	0.001
Hodgkin lymphoma	2.22	1.83–2.67	0.001
Follicular and B-cell lymphoma	1.86	1.77–1.95	0.001
T/NK-cell lymphomas	2.14	1.77–2.57	0.001
Multiple myeloma	1.49	1.39–1.59	0.001
Leukemias	1.82	1.74–1.91	0.001
Chronic myeloproliferative disorders	1.11	1.03–1.2	0.001
Comorbidities			
Solid tumor	2.33	2.29–2.38	0.001
Diabetes	1.01	1–1.02	0.15
Coronary disease	1.09	1.07–1.11	0.001
Heart failure	1.07	1.05–1.08	0.001
Hypertension	0.93	0.92–0.95	0.001
Obesity	0.96	0.94–0.98	0.001
Dementia	1.24	1.22–1.27	0.001
Cerebrovascular disease	2.06	1.99–2.14	0.001
Chronic liver disease	1.6	1.5–1.7	0.001
Chronic kidney disease	1.14	1.12–1.16	0.001
Chronic pulmonary disease	0.71	0.7–0.73	0.001

**Table 5 jcm-13-05400-t005:** Multivariate analyses of the cohort with hematological patients only (binary logistic regression).

	OR (Adjusted)	95%CI	*p* Value
Sex (men)	1.12	1.06–1.19	0.001
Age	1.03	1.02–1.03	0.001
Length of hospital stay	0.97	0.97–0.98	0.001
ICU admission	6.68	6.15–7.26	0.001
Length of ICU stay	1.03	1.03–1.04	0.001
Type of malignancy			
Hodgkin lymphoma	1.58	1.25–1.98	0.001
Follicular and B-cell lymphoma	1.52	1.31–1.77	0.001
T/NK-cell lymphomas	1.68	1.33–2.1	0.001
Multiple myeloma	1.24	1.06–1.45	0.01
Leukemias	1.6	1.39–1.84	0.001
Chronic myeloproliferative disorders	1.05	0.92–1.2	0.48
Comorbidities			
Solid tumor	1.55	1.4–1.72	0.001
Diabetes	0.92	0.87–0.99	0.02
Coronary disease	1.05	0.96–1.15	0.3
Heart failure	1.25	1.17–1.35	0.001
Hypertension	1.03	0.96–1.09	0.4
Obesity	0.97	0.87–1.08	0.55
Dementia	1.12	0.99–1.27	0.07
Cerebrovascular disease	2.04	1.59–2.62	0.001
Chronic liver disease	1.28	0.94–1.73	0.11
Chronic kidney disease	1.15	1.07–1.24	0.001
Chronic pulmonary disease	0.73	0.67–0.79	0.001

## Data Availability

A contract signed with the Spanish Health Ministry, which provided the dataset, prohibits the authors from providing their data to any other researcher. Furthermore, the authors must destroy the database upon the conclusion of their investigation. The database cannot be uploaded to any public repository. However, we uploaded Python code and some pieces of the whole dataset to a public repository at https://github.com/rafalinux/covid-hematological.

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
