# Peer review of "Outcomes and Patterns of Evolution of Patients with Hematological Malignancies during the COVID-19 Pandemic: A Nationwide Study (2020–2022)"

_jcm, 2024, doi:10.3390/jcm13185400_

Round 1

Reviewer 1 Report

Comments and Suggestions for Authors

This study conducted in Spain found that hematological malignancy (HM) is a significant risk factor for morbidity and mortality in COVID-19 patients. The study involved 34,962 patients with HM and found that HM was a significant risk factor for mortality. The overall mortality rate was higher for patients with HM, lymphomas, multiple myelomas, and leukemias. Non-Hodgkin lymphomas were the highest risk factor for mortality. The study highlights the need for special prophylactic and therapeutic measures for HM patients. Prior to making a decision on publication, it is crucial to address specific uncertainties that require clarification.

 1.     Patients with hematological malignancies face a heightened risk of mortality from COVID-19 and are more prone to experiencing poor prognoses. Nevertheless, the study encompasses the years 2020 to 2022, during which the COVID-19 vaccination is expected to play a significant role in building immunity against the virus among the general population. Patients with HM may not be eligible for COVID-19 inoculation.

2.     Line 65: The authors utilized coding for COVID-19 in all diagnostic positions during data collection. However, some patients experienced subclinical symptoms and may have been hesitant to seek medical attention.

3.     Line 118~119: “For individuals with HM, we calculate an estimated ratio for hospitalized patients of 5.87% per year, that is, greater than the general population.” The conclusion appears to be drawn in a rather arbitrary manner. Patients with HM may need to be hospitalized if they show signs of COVID-19 infection.  

4.     Line 238~239: Patients diagnosed with leukemia and myelodysplastic syndrome experienced the highest mortality rate. It would be helpful if the authors could clarify whether the patients in question are receiving more intense chemotherapy compared to those with non-Hodgkin lymphoma and plasma cell disorders.

5.     The conclusion could benefit from more solid support and additional details to strengthen the argument.

Comments on the Quality of English Language

The English language proficiency is satisfactory.

Author Response

Comment 1:

Patients with hematological malignancies face a heightened risk of mortality from COVID-19 and are more prone to experiencing poor prognoses. Nevertheless, the study encompasses the years 2020 to 2022, during which the COVID-19 vaccination is expected to play a significant role in building immunity against the virus among the general population. Patients with HM may not be eligible for COVID-19 inoculation.

Response 1:

Thank you for pointing this out. We agree with your comment. The beginning of vaccination rollout in patients with malignancies in Spain in March 2021, including patients with HM. Unfortunately, we have no data regarding the role of vaccination in patients with HM, and discussion on this topic is purely speculative.

Comment 2:

The authors utilized coding for COVID-19 in all diagnostic positions during data collection. However, some patients experienced subclinical symptoms and may have been hesitant to seek medical attention.

Response 2:

In our retrospective study we used using data extracted from hospitalized patients only, exclusively constructed from discharge reports. All includes patients were hospitalized. It is important to note that patients with HM tend to have frequent medical visits, and that not all visits are due to symptoms related to COVID-19, but all hospitalized patients with HM had COVID-19. While this issue may result in bias, we consider that, on one hand, overestimation is unlikely, due to the large number of patients included, and, on the other hand, COVID-19, although mild presentation, can destabilize a chronic condition, such a hematological malignancy.

Comment 3:

“For individuals with HM, we calculate an estimated ratio for hospitalized patients of 5.87% per year, that is, greater than the general population.” The conclusion appears to be drawn in a rather arbitrary manner.

Response 3:

We apologize if we did not explain ourselves. There is no arbitrary calculation. The Spanish Oncology Society estimated there were 198,507 patients with HM in Spain in 2022, being 595,521 cummulated patients with HM in the period 2020-2022. In this period we identified 34,962 hospitalized patients with HM. That is, 34,962 ÷ 595,521 = 5.87%.

Comment 4:

Line 238~239: Patients diagnosed with leukemia and myelodysplastic syndrome experienced the highest mortality rate. It would be helpful if the authors could clarify whether the patients in question are receiving more intense chemotherapy compared to those with non-Hodgkin lymphoma and plasma cell disorders.

Response 4:

Thank you for pointing this out. We agree with your comment. Unfortunately, the main limitation of our manuscript was the use of an administrative database. Some clinical data were not available, such as vaccination and medications. Therefore, we could not assess the effect of certain treatments, such as specific anti-cancer treatments or immunotherapies. Since we have no data regarding the role of chemotherapy in patients with HM, the discussion on this topic is purely speculative.

Comment 5:

The conclusion could benefit from more solid support and additional details to strengthen the argument.

Response 5:

We have re-written the conclusions:

“This study presents the first comprehensive, nationwide analysis of the epidemiology and risk factors for in-hospital mortality among patients with hematological malignancies (HM) and COVID-19 in Spain. It highlights the unique vulnerabilities of this population. Our findings indicate a significantly higher hospitalization rate (5.8%) for patients with HM compared to the general population (0.84%), and an overall mortality rate of 19.8%.

We found discrepancies in mortality rates between our research and other studies, potentially due to differences in study populations, observation periods, and pandemic phases. Notably, the mortality rate has declined over time, possibly reflecting improved management and vaccination, although we are aware that specific data on vaccination were not available in this study.

Our research highlights he need for targeted prophylactic and therapeutic interventions. Close monitoring and tailored care are essential to mitigate the severe outcomes of COVID-19 in this high-risk population. Our study provides valuable insights into the impact of COVID-19 on patients with HM in a population-based setting. Our findings emphasize the importance of continued vigilance and tailored healthcare strategies for at-risk groups in the ongoing management of COVID-19.”

Reviewer 2 Report

Comments and Suggestions for Authors

I would like to thank the editors of the journal for the opportunity to review the manuscript on the population of patients with hematological malignancies during the COVID-19 pandemic. This is an impressive work, which is the result of a national investigation conducted during the period 2020-2022, and which included close to 35 thousand patients with hematological malignancies. The study analyzed the impact of COVID-19 infection on the course of the disease and the survival of patients with hematological malignancies, i.e. on the risk of death in the respective subgroups of patients. It was indicated that the population of patients with hematological malignancies is very vulnerable, and that non-Hodgkin's lymphoma is the most significant risk factor for mortality, followed by leukemia, Hodgkin's lymphomas and blood dyscrasias.

·        Reviewer suggestions:

·        • The methodological part of the manuscript is written in detail, it is only necessary to indicate in the text the number of the decision of the ethical committee that approved the conducted research.

·        • In the description of statistical methods, the authors state that continuous variables are presented as median with interquartile range. Does this mean that the distribution of all variables was not normal, considering that in the case of a normal distribution, instead of the median, the mean value with the standard deviation is stated?

·        state the statistical test that obtained the corresponding p values ​​listed in the table (under Table 1 and 2)

·        Add to table 5 the data of multivariate regression analysis related to the influence of length of hospitalization, ICU administration and ICU length of stay on the outcome

·        he conducted research makes a significant contribution to the understanding of predictors of poor outcome with people suffering from hematological malignancies, which enables the creation of special prevention measures and the individualization of procedures in the treatment of people with this type of neoplasm.

Author Response

Comment 1:

The methodological part of the manuscript is written in detail, it is only necessary to indicate in the text the number of the decision of the ethical committee that approved the conducted research.

Response 1:

We included the information regarding the approval of the Ethical Committee in line 342: “This study was approved by the Ethical Board of Universidad Rey Juan Carlos (ID number 2610202334423)”.

Comment 2:

In the description of statistical methods, the authors state that continuous variables are presented as median with interquartile range. Does this mean that the distribution of all variables was not normal, considering that in the case of a normal distribution, instead of the median, the mean value with the standard deviation is stated?

Response 2:

Thank you for pointing this out. We agree with your comment. We identified three continuous variables (age, length of hospital stay and length of ICU stay). All these variables were checked for normality and we observed that data were not normally distributed. That was the reason we used median and interquartile range instead of mean and standard deviation.

Comment 3:

Please, state the statistical test that obtained the corresponding p values listed in the table (under Table 1 and 2).

Response 3:

We added “Chi-square test for continuous variables and Mann–Whitney U test for categorical variables were performed as tests of independence” under Tables 1 and 2.

Comment 4:

Please, add to table 5 the data of multivariate regression analysis related to the influence of length of hospitalization, ICU admission and ICU length of stay on the outcome.

Response 4:

We have added the rows regarding the length of stay and ICU admission to Table 5, as requested.

Reviewer 3 Report

Comments and Suggestions for Authors

This is an interesting manuscript of epidemiology in patients with haematological malignancies and COVID-19. Study was performed from 2020 to 2022. It would be interesting, in order to improve the manuscript, if authors added a comparison  between patients that received vaccine and the ones that did not , or even the time post vaccination in which they were. 

The kind of treatment anti-cancer probably would bring some interesting discussion too.

Analysing Figure A1, I ask the authors what could be the reason for the increase in hopital admission in the beggining of 2020 (beggining of COVID-19) but also the beggining of 2022. 

Author Response

Comments 1 and 2:

It would be interesting, in order to improve the manuscript, if authors added a comparison between patients that received vaccine and the ones that did not , or even the time post vaccination in which they were. The kind of treatment anti-cancer probably would bring some interesting discussion too.

Responses 1 and 2:

Thank you for pointing this out. We agree with your comment. Unfortunately, the main limitation of our manuscript was the use of an administrative database. Some clinical data were not available, such as vaccination and medications. Therefore, we could not assess the effect of immunization or certain treatments, such as specific anti-cancer treatments or immunotherapies. Since we have no data regarding the role of vaccination in patients with HM, the discussion on this topic is purely speculative.

Comment 3:

Analyzing Figure A1, I ask the authors what could be the reason for the increase in hospital admission in the beginning of 2020 (beginning of COVID-19) but also the beginning of 2022.

Response 3:

As pointed out in https://doi.org/10.1186/s12879-023-08454-y, and regarding the distribution of waves over time, they can hypothesize only about the association, bu not causation, between certain events and peaks of hospitalizations in Spain. So, the peak in December 2021 showed an increase in hospitalizations, but not in mortality, showing the beneficial effect of vaccination. But regardless of vaccination, it seems that waves and peaks were related to social events: holidays, Christmas gatherings, and the relaxation of public health measures such as social distancing.
